# NNETSCAPE NAVIGATOR: COMPLEX DEMONSTRATIONS FOR WEB AGENTS WITHOUT A DEMONSTRATOR

## ABSTRACT

We introduce NNetscape Navigator (`NNetnav`), a method for training web agents entirely through synthetic demonstrations. These demonstrations are collected by first interacting with a browser to generate trajectory rollouts, which are then retroactively labeled into instructions using a language model. Most work on training browser agents has relied on expensive human supervision, and the limited previous work on such *interaction-first* synthetic data techniques has failed to provide effective search through the exponential space of exploration. In contrast, `NNetnav` exploits the hierarchical structure of language instructions to make this search more tractable: complex instructions are typically decomposable into simpler subtasks, allowing `NNetnav` to automatically prune interaction episodes when an intermediate trajectory cannot be annotated with a meaningful sub-task. We use `NNetnav` demonstrations from a language model for supervised fine-tuning of a smaller language model policy, and find improvements of 6 points on WebArena and over 20 points on MiniWoB++, two popular environments for web-agents. Notably, on WebArena, we observe that language model policies can be further enhanced when fine-tuned with `NNetnav` demonstrations derived from the *same* language model. Finally, we collect and release a dataset of over 6k `NNetnav` demonstrations on WebArena, spanning a diverse and complex set of instructions.

## 1 INTRODUCTION

Building grounded language agents that map human language instructions to a sequence of executable actions is a long-standing goal of artificial intelligence (Winograd, 1972), with the ultimate goal of automating mundane web tasks like flight booking. A promising new approach for building such agents is to use large language models to control policies in digital environments like browsers (Yao et al., 2022; Shinn et al., 2023; Murty et al., 2024; Wang et al., 2024, among others).

Unfortunately, such *grounded instruction following* without any training examples is challenging because LMs do not know about the myriad and ever changing interaction possibilities of different websites. For instance, for a new online shopping website, a zero-shot LM agent may not know how to make a return or change order details, without expensive test-time exploration. Even simple tasks like selecting a flight might involve typing in airport codes or selecting from a drop-down menu, and zero-shot agents cannot know this a-priori.

One way to provide LM web-agents with knowledge about new web interfaces is via expert demonstrations, that can either be used for in-context learning (Yao et al., 2022) or supervised fine-tuning (Lai et al., 2024). These demonstrations are either fully provided by human experts (Sodhi et al., 2023; Yao et al., 2022) or consist of human-generated trajectories paired with model-generated instructions (Lai et al., 2024). Of course, collecting human demonstrations that cover each possible use case for every web-site is an unattractively large, never-ending task. Thus, recent work uses entirely synthetic demonstrations by sampling a synthetic instruction, and then mapping it into a trajectory using a base LLM agent (Patel et al., 2024; Murty et al., 2024).

Such *instruction-first* methods for data collection face several challenges. First, synthetic instructions in these demonstrations are sampled from an ungrounded LM prior that generates only plausible[1]

---

[1] We use the term *plausible* for instructions that match a website's genre or intended use. For example, searching for clothes on a retail site or checking notifications on a social media platform. Not all plausible instructions are feasible.

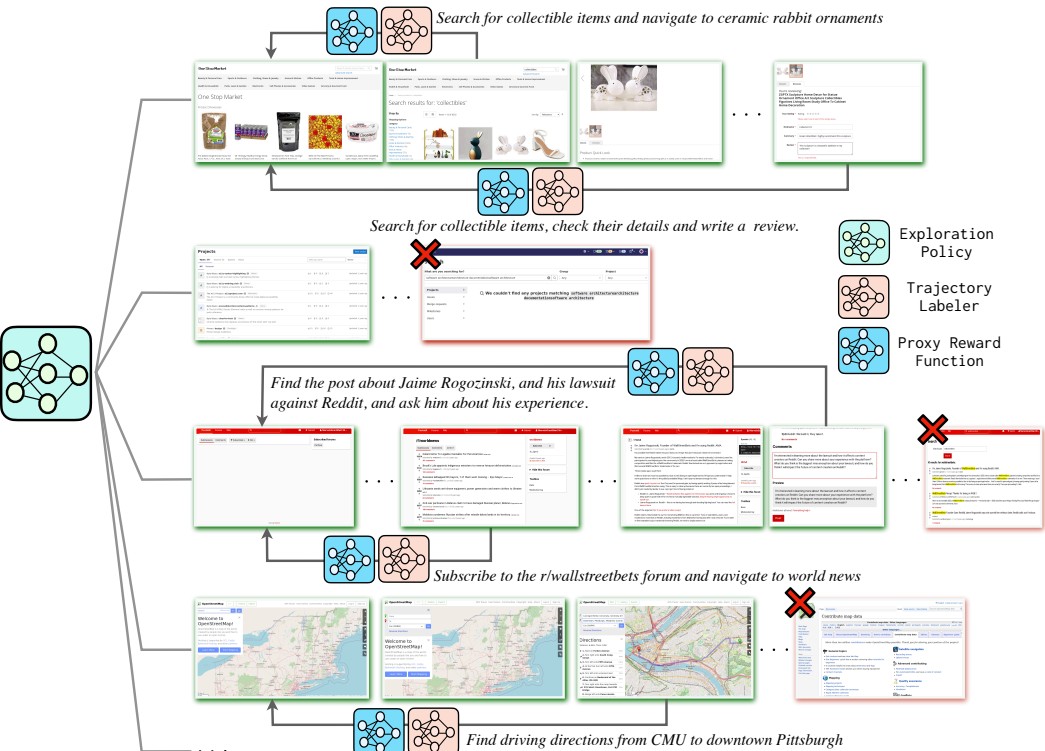

Figure 1: `NNetnav` produces synthetic demonstrations for training web-agents by exploring a website to create trajectories, and then labeling them into instructions. Long exploration in `NNetnav` is made efficient using a pruning heuristic inspired by the hierarchical structure of complex instructions. At fixed intervals during exploration, the labeling function infers an instruction for the trajectory so far, and if the resulting (instruction, trajectory) pair receives a low score from a reward function, the episode is terminated (red cross). Components in `NNetnav` are implemented using prompts to the same LM.

instructions without ensuring feasibility; *e.g.*, an instruction such as *Respond to the first post on r/callofdutyfans* for reddit is plausible, but not always feasible. Second, generated instructions are limited to those that reference visible features of the website; *e.g.*, given the landing page of a github-like platform, no LM prior can generate instructions like *Find information about Eric Bailey's contributions to the byteblaze project*, which requires knowing about deeply embedded website-specific entities like *Eric Bailey*. Finally, these methods provide no control over the complexity of instructions, and rely entirely on the LM or bespoke prompts to generate complex instructions.

Instead of starting with a sampled instruction, we start by sampling an *interaction* first, and then retroactively labeling it into an instruction that is feasible by design. At a high-level, our approach *NNetscape Navigator* (`NNetnav`, Fig 1), uses a language model exploration policy to perform extended interactions with an environment, and another language model trajectory labeler to annotate trajectories with instructions[2]. To effectively control the exponential space of meaningful interactions, `NNetnav` uses the hierarchical structure of language instructions as a pruning heuristic: for exploration to discover a meaningfully complex task, trajectory prefixes must correspond to meaningful sub-tasks. Thus, during an exploration episode, if a language model cannot label trajectory prefixes (at set time-steps) with a sub-task, further exploration is automatically terminated. Imposing such a structure over search not only enhances efficiency, but also results in complex and hierarchical instructions (See Table 7 for examples). `NNetnav` prompts the same base language model for exploration, relabeling and inferring sub-tasks, and effectively addresses all limitations of instruction-first data collection.

---

[2]We will open-source our code upon acceptance.

Using `GPT-4o-mini` (Achiam et al., 2023) as our base language model, we use demonstrations collected via `NNetnav` to fine-tune a smaller `Llama-3-8B-Instruct` (Dubey et al., 2024) based agent on two benchmarks for web navigation, MiniWoB++ (Shi et al., 2017; Liu et al., 2018) and WebArena (Zhou et al., 2023). Compared to the base agent, performance of the fine-tuned agent improves from 28% to 48% on MiniWoB++, and from 1% to 7% on WebArena. Crucially, these improvements are higher than those from a model that's fine-tuned with an instruction-first data collection method. Finally, we find that `NNetnav` can be used for self-training—fine-tuning a small LM agent with `NNetnav` demonstrations from the *same* LM leads to an improvement of 4% points absolute (1% to 5%) on WebArena. Further analysis reveals the benefits of retroactive labeling beyond performance improvements: When using a model-based evaluator, similar to Pan et al. (2024), hindsight trajectories from `NNetnav` have a higher mean reward than trajectories from an LLM agent based on the same underlying language model. Finally, we collect and release `NNetnav-6k`, a dataset of over 6k demonstrations covering a wide and complex range of use cases on WebArena.

## 2 BACKGROUND

Following instructions on web-browsers is a deterministic sequential decision making problem. Given an instruction $g$, an instruction following agent interacts with the browser by issuing a sequence of actions $\langle a_1, a_2, \ldots, a_T \rangle$ where each $a_i \in \mathcal{A}$ is drawn in response to an observation $o_i$. Executing an action causes a state transition based on some unknown but deterministic environment dynamics, leading to a new observation $o_{i+1}$. The entire episode can be summarized as a trajectory $\tau \coloneqq \langle o_1, a_1, o_2, a_2, \ldots o_T, a_T, o_{T+1} \rangle$. We formalize the instruction following agent as a mapping $\pi(a_t \mid \tau_{<t}; g)$ where $\tau_{<t} \coloneqq \langle o_1, a_1, \ldots o_t \rangle$ is the trajectory so far. In this framework, the action space $\mathcal{A}$ consists of a finite set of strings, while observations are represented as either flattened DOM trees or website accessibility trees.

**LLMs as Instruction Following Agents.** Recent work explores prompted large language models (LLMs) to directly parameterize $\pi$. These methods typically work in settings with textual observations and action spaces, and many output a reasoning string $r_i$ before predicting the action string $a_i$. Concretely, we formalize an LM agent (omitting prompts) as $\pi_{\text{LM}}(a_t \mid \tau_{<t}; g) \coloneqq p_{\text{LM}}(a_t, \mid \tau_{<t}, r_t; g)$ where $r_t \sim p_{\text{LM}}(r \mid \tau_{<t}; g)$ is a reasoning step drawn as a sample from the LM.

Given expert demonstrations $\{g^i, \tau^i\}$ where $\tau^i \coloneqq \langle o_1^i, r_1^i, a_1^i, o_2^i, r_2^i, a_2^i \ldots o_T^i \rangle$, prior work adapts LM agents using demonstrations either as in-context examples (Yao et al., 2022; Shinn et al., 2023; Sun et al., 2023; Kim et al., 2023, among others) or as training data for supervised fine-tuning (Furuta et al., 2023; Lai et al., 2024; Lù et al., 2024; Patel et al., 2024). For supervised fine-tuning of $\pi_{\text{LM}}$ on a dataset of demonstrations, we construct training instances $\{(g^i, \tau_{<t}^i), (r_t^i, a_t^i)\}$ where $r_t^i, a_t^i$ serves as the target reasoning step and action for an intermediate context $(g^i, \tau_{<t}^i)$.

**Data collection with instruction-first methods.** Collecting human demonstrations for training web-agents is time consuming and costly. Thus, recent work proposes methods for generating synthetic data for web-agents using language model components (Lai et al., 2024; Furuta et al., 2023; Murty et al., 2024). These methods start by sampling synthetic instructions from an instruction generator (a prompted LM that takes the website landing page and a persona as input), and then use a zero-shot LM policy to convert these instructions into trajectories. Resulting demonstrations are filtered using either the ground truth reward function (Furuta et al., 2023), or using another language model based reward function (Lai et al., 2024; Murty et al., 2024). Most of these methods use bigger and better language models for collecting demonstrations, and then use this data to adapt smaller models.

## 3 OUR APPROACH

`NNetnav` (Fig 1) is an *interaction-first* method for constructing demonstrations: An exploration policy interacts with a browser in a structured manner to sample long trajectories which are retroactively labeled into instructions (§3.2). We then post-process each trajectory to add post-hoc reasoning steps corresponding to the generated instructions, and then use this data for supervised fine-tuning (§3.3). We provide detailed pseudo-code for the exploration and relabeling steps in `NNetnav` in Algorithm 1.

---

**Algorithm 1:** Exploration and Relabeling in `NNetnav` within a single interaction episode

---

**Input**: $\pi_{\text{explore}}, \text{Lf}_{\text{LM}}, s_{\text{LM}}, \Delta_{\text{LM}}$

**Function** `run_retroactive_labeler(`$\tau$`)`:
$\quad \delta_\tau \leftarrow [\Delta_{\text{LM}}(o_t, a_t, o_{t+1}) \mid (o_t, a_t, o_{t+1}) \in \tau]$;
$\quad g_{\text{curr}} \leftarrow \text{Lf}_{\text{LM}}(\delta_\tau)$;
$\quad r_{\text{curr}} \leftarrow s_{\text{LM}}(i_{\text{curr}}, \delta_\tau)$;
$\quad$**return** $g_{\text{curr}}, r_{\text{curr}}$;

**Function** `explore(`$\mathcal{W}, T_{prune}$`)`:
$\quad o_1 \leftarrow \mathcal{W}$.init-observation;
$\quad t \leftarrow 1$;
$\quad \tau \leftarrow \langle \rangle$;
$\quad$demonstrations $\leftarrow []$;
$\quad$**while** $t \leq T_{max}$ **do**
$\quad\quad$**if** $t \in T_{prune}$ **then**
$\quad\quad\quad g_{\text{curr}}, r_{\text{curr}} \leftarrow$ `run_retroactive_labeler(`$\tau$`)`;
$\quad\quad\quad$**if** $r_{curr} < 1$ **then**
$\quad\quad\quad\quad$**break**;
$\quad\quad\quad$**else**
$\quad\quad\quad\quad$demonstrations.add($(g_{\text{curr}}, r_{\text{curr}})$);

$\quad\quad a_t \leftarrow \pi_{\text{explore}}(o_t)$;
$\quad\quad o_{t+1} \leftarrow \mathcal{W}$.step($a_t$);
$\quad\quad \tau$.add($(o_t, a_t, o_{t+1})$);
$\quad\quad t \leftarrow t + 1$;

$\quad$**return** demonstrations;

---

## 3.1 LM Components

We start by describing various components in `NNetnav`. All of these components are implemented by zero-shot prompting a language model, with different prompts (see Appendix A for details).

**Exploration Policy.** To interact with the environment, we use an exploration policy $\pi_{\text{explore}}$, implemented using a chain-of-thought prompted language model (Wei et al., 2022). Additionally, to simulate a diverse set of behaviors from users and improve the diversity of resulting trajectories, we seed each episode with a string description of a plausible user persona for the given website (Shanahan et al., 2023; Argyle et al., 2023).

**Summarizing Trajectory changes.** Actions issued by $\pi_{\text{explore}}$ result in a new observation in the environment. We summarize this change as a short string description via another module $\Delta_{\text{LM}}$, implemented using a language model. In particular, for any state $o_t$, action $a_t$ and the resulting next state $o_{t+1}$, $\delta_t = \Delta_{\text{LM}}(o_t, a_t, o_{t+1})$ produces a string-valued description of the changes in the observation as a result of the action. For a trajectory $\tau$, we denote the sequence of state changes as $\delta_\tau$

**Trajectory Labeling Function.** Given state changes $\delta_\tau$, a labeling function $\text{Lf}_{\text{LM}}$ produces a plausible instruction $\hat{g} = \text{Lf}_{\text{LM}}(\delta_\tau)$ that the agent could have followed to produce the given interaction.

**Reward Function.** Given $\hat{g}$ and $\delta_\tau$, the reward function module assigns a reward $s_{\text{LM}}(\hat{g}, \delta_\tau) \in \{0, 1\}$, based on how well the state changes correspond to the given instruction $\hat{g}$.

## 3.2 Sampling Demonstrations via Interactions

Let $t_{\max}$ be the maximum episode length for each exploration rollout. At specific time-steps $\{t_1, t_2, \ldots t_{\max}\}$, we run the pruning heuristic that attempts to annotate the trajectory so-far with a sub-task annotation. If this is successful, we continue the episode, and otherwise halt to sample another rollout. Concretely, suppose we have a partial trajectory $\tau_{<t}$ after interacting with the environment for $t$ time-steps. If $t \in \{t_1, t_2, \ldots t_{\max}\}$, we first obtain a retroactive sub-task $\hat{g} = \text{Lf}_{\text{LM}}(\delta_{\tau_{<t}})$. We halt further exploration if $s_{\text{LM}}(\hat{g}, \delta_{\tau_{<t}}) = 0$. Otherwise, we add the generated $(\hat{g}, \tau_{<t})$ to our set

of synthetic demonstrations, and continue exploring. Typically, each interaction episode results in multiple demonstrations.

### 3.3 GENERATING POST-HOC REASONING STEPS

The exploration policy in this work is implemented using a language model that generates a reasoning step, before choosing an action (§2). Since actions in our demonstration set are a result of exploration, corresponding reasoning steps are not generally related to the retroactively generated instruction. Thus, for each demonstration in our synthetic demonstration set, we post-hoc annotate every action with a new reasoning step that directly corresponds to the generated instruction. Concretely, given every $(\hat{g}, o_i, a_i)$ tuple in our synthetic demonstration set, we prompt a language model to output a suitable reasoning step for choosing action $a_i$ given the instruction $\hat{g}$ and current observation $o_i$. We note that such a post-hoc reasoning procedure is similar to Yang et al. (2024).

## 4 EXPERIMENTAL SETUP

### 4.1 DATASETS

We fine-tune language model policies with `NNetnav` demonstrations on two web navigation environments, MiniWoB++ (Shi et al., 2017; Liu et al., 2018) and WebArena (Zhou et al., 2023).

1. **MiniWoB++:** A dataset of diverse synthetic web-interfaces with a shared action space. Tasks on MiniWoB++ range from clicking on buttons to complex tasks like making a booking on a website. We use a subset of 8 complex tasks from MiniWoB++ as a toy benchmark to evaluate our method. We use the `bid`-based action space from BrowserGym (Drouin et al., 2024), consisting of 12 actions, and a DOM based observation space. Due to its synthetic nature, MiniWoB++ comes with an automatic reward function. We report the mean reward over 20 random seeds for each task, similar to (Drouin et al., 2024).

2. **WebArena:** A dataset of realistic web navigation tasks over 5 websites covering domains such as e-commerce, discussion forums, maps and software development. We use the default action space from WebArena (typing, clicking, hovering, tab management) and the default accessibility tree based observation space. WebArena consists of 812 Web navigation tasks across these websites, and provides an evaluator that measures success rate (SR) in terms of functional correctness. We report the average SR across these tasks.

### 4.2 MODEL SETTINGS

All inference evaluations are conducted using the same base language model, with data collection typically performed using a larger language model unless stated otherwise. We evaluate under the following settings:

1. **Zero-Shot:** A baseline zero-shot LM policy $\pi_{\text{LM}}$, prompted using chain-of-thought prompting (Wei et al., 2022). Next, we consider various fine-tuned models.

2. **SFT (Instruction-First):** Supervised fine-tuning (SFT) of the base policy using data collected via instruction-first sampling. Here, we use the same reward model for filtering demonstrations as `NNetnav`, and also sample the same number of demonstrations for fair comparison.

3. **SFT (`NNetnav`):** Supervised fine-tuning of $\pi_{\text{LM}}$ with demonstrations collected via `NNetnav`.

4. **SFT (`NNetnav` + Distil.):** Ablation, where we only retain instructions found via `NNetnav` and re-generate trajectories by prompting the same large LM as an agent. We use this setting to isolate performance improvements attributable to `NNetnav` trajectories.

### 4.3 IMPLEMENTATION DETAILS

All LM components for data collection in `NNetnav` as well as instruction-first methods are based on `GPT-4o-mini` (specifically `gpt-4o-mini-2024-07-18`). We use `Llama-3-8B-Instruct` as the inference policy $\pi_{\text{LM}}$. For Instruction-first data collection, we sample 50 instructions per

| Domain | Zero-Shot | SFT (Instruction-First) | SFT (`NNetnav`) | SFT (`NNetnav` + Distil.) |
|---|---|---|---|---|
| *MiniWoB++* | | | | |
| book-flight | 0.0 | 0.0 | 0.0 | 0.0 |
| choose-date | 0.0 | 0.0 | 0.0 | 0.0 |
| click-checkboxes-soft | 0.4 | 0.25 | **0.65** | 0.5 |
| email-inbox | 0.25 | 0.3 | 0.3 | **0.35** |
| login-user | 0.3 | 0.0 | **1.0** | 0.95 |
| navigate-tree | 1.0 | 0.95 | **1.0** | 0.95 |
| phone-book | 0.15 | 0.15 | 0.2 | **0.55** |
| use-autocomplete | 0.25 | 0.55 | **0.7** | 0.35 |
| Avg. | 0.28 | 0.28 | **0.48** | 0.36 |
| *WebArena* | | | | |
| Shopping | 3.8 | **7.7** | 7.4 | 7.4 |
| CMS | 0 | 4.2 | 4.2 | 4.2 |
| Reddit | 0 | 0 | 0 | 0 |
| Gitlab | 0 | 0 | 0 | 4.5 |
| Maps | 0 | 9.1 | **28.5** | 15.4 |
| Avg. | 1.0 | 4.2 | **7.2** | 6.0 |

Table 1: Results for MiniWoB++ and WebArena, broken down by domain, reporting mean reward for MiniWoB++ and task success rate (SR) for WebArena. We compare the zero-shot agent with agents fine-tuned with `NNetnav` and instruction-first demonstrations. Overall, fine-tuning with `NNetnav` leads to the largest improvements: from 28% to 48% on MiniWoB++; from 1% to 7.2% on WebArena.

website for WebArena, and 80 instructions per interface in MiniWoB++, and prompt the instruction generator with the landing page as well as a persona (to improve diversity). For `NNetnav`, we use our exploration policy to generate 50 episodes per website for WebArena, and 80 episodes per interface for MiniWoB++. We set $T_{max}$ to 40 for WebArena, and 20 for MiniWoB++. For both MiniWoB++ and WebArena, we apply the pruning function every 4 time-steps. We use 16 persona types per website for WebArena, and 10 persona types per web-interface for MiniWoB++.

We use the BrowserGym framework (Drouin et al., 2024) for experiments with MiniWoB++ and prune out the full DOM to only keep visible elements. During inference, we set the max episode length for $\pi_{LM}$ as 30 for WebArena (following Zhou et al. (2023)), and 20 for MiniWoB++. We-bArena requires agents to output a special `stop` action for outputting answers. We post-process `NNetnav` demonstrations to add a `stop` action at the end of the trajectory using a prompted LM (See Appendix A.2 for details).

**Fine-tuning details.** We fine-tune all models for 5 epochs, truncating the max sequence length to 4096, with a learning rate of 2e-5, using 4 A100 GPUs. We provide complete details of our training setup in Appendix D. We use open-instruct (Wang et al., 2023) for fine-tuning all language models, and set up local inference servers using vllm (Kwon et al., 2023).

## 5 MAIN RESULTS

**Fine-tuning agents with `NNetnav` leads to consistent gains.** We report results from all model settings in Table 1. We find that fine-tuning the zero-shot policy $\pi_{LM}$ with synthetic demonstrations from `NNetnav` leads to consistent improvements on all tasks in MiniWoB++, leading to a 20 point improvement overall. We note an improvement of over 6 points from fine-tuning with `NNetnav` demonstrations on WebArena. Importantly, gains from fine-tuning with `NNetnav` exceeds those from using instruction-first methods by 12 points on MiniWoB++ and 1.2 points on WebArena.

| Domain | Zero-Shot | SFT (Instruction-First) | SFT (NNetnav) | SFT (NNetnav + Distil.) |
|---|---|---|---|---|
| Shopping | 1.26 | 1.37 | 2.22 | **2.33** |
| CMS | 1.21 | 1.29 | **1.92** | 1.87 |
| Reddit | 1.08 | 1.31 | **2.0** | 1.54 |
| Gitlab | 1.14 | 1.09 | **1.86** | 1.5 |
| Maps | 1.21 | 1.36 | **2.29** | 1.86 |
| WebArena (Avg.) | 1.19 | 1.28 | **2.05** | 1.87 |

Table 2: Model-based evaluation on WebArena, broken down by domain. For each test instruction and predicted trajectory, we prompt a `GPT-4o` based reward model to output a graded reward from 1 to 5 based on a manual rubric. We find that fine-tuning with `NNetnav` outperforms all other settings.

| Domain | Zero-Shot | Self-Train (NNetnav) |
|---|---|---|
| Shopping | 3.8 | **15.4** |
| CMS | 0.0 | 0.0 |
| Reddit | 0.0 | 0.0 |
| Gitlab | 0.0 | 0.0 |
| Maps | 0.0 | **7.1** |
| Avg. | 1.0 | **5.3** |

Table 3: We generate `NNetnav` demonstrations using `Llama-3-8B-Instruct`, which we use for supervised fine-tuning of an agent based on the same LM, and find significant improvements on WebArena from 1% to 5.3%.

| Domain | Zero-Shot | Self-Train (NNetnav) |
|---|---|---|
| *in-domain* | | |
| Shopping | 1.26 | **1.37** |
| CMS | 1.21 | **1.29** |
| Maps | 1.21 | **1.36** |
| *out-of-domain* | | |
| Reddit | 1.08 | **1.31** |
| Gitlab | **1.14** | 1.09 |

Table 4: We fine-tune $\pi_{\text{LM}}$ with `NNetnav` demonstrations from 3 websites, and evaluate in-domain and out-of-domain generalization with the model based evaluator that outputs a score from 1 to 5. While improvements are higher in-domain, we still find improvements on out-of-domain data.

**Fine-grained evaluation on WebArena with LLM reward.** We observe highly non-uniform improvements on WebArena with no improvements on Reddit and Gitlab in particular. We hypothesize that this is due to the coarse nature of WebArena's success rate (SR) evaluation, since it does not provide partial credit. Thus, inspired by Pan et al. (2024), we develop a model based evaluation using the largest publicly available `GPT-4o` (specifically `gpt-4o-2024-08-06`) model to assign a graded reward from 1 to 5 to model outputs for each test instruction (see Appendix B for full prompt). We present results from model-based evaluation in Table 2. At the level of model settings we observe the same trend: Zero-Shot < SFT (Instruction-First) < SFT (NNetnav + Distil.) < SFT (NNetnav). However since this evaluation is more graded, we find consistent improvements from using `NNetnav` demonstrations across all websites, including Reddit and Gitlab, where improvements of 0.92 points and 0.72 points are observed, respectively. As expected, performance rankings sometimes changes with such graded evaluation *e.g.* on CMS, all fine-tuned models are tied in terms of SR (Table 1), but not in terms of graded reward (Table 2). Overall, we believe WebArena evaluations should incorporate both overall SR and fine-grained model based evaluation for a more comprehensive understanding of system performance.

**The Benefit of Hindsight.** We find that SFT (NNetnav) outperforms SFT (NNetnav + Distil.) on both MiniWoB++ and WebArena. Trajectories in `NNetnav` are obtained via a hindsight procedure: the model acts *first*, and the instruction is inferred afterward. In constrast, for `NNetnav + Distil.`, the instruction is provided first, and the trajectory is sampled later. To understand if hindsight trajectories offer an advantange, we use the model based evaluator to measure training data quality for these settings. Specifically, we use the evaluator to assign reward to trajectories in `NNetnav` and `NNetnav + Distil.` for WebArena, and find a win-rate of 64% for `NNetnav` trajectories with a mean reward of 3.52 compared to a reward of 2.44 for `NNetnav + Distil.` We conclude that gains from `NNetnav` extend beyond just providing more complex instructions.

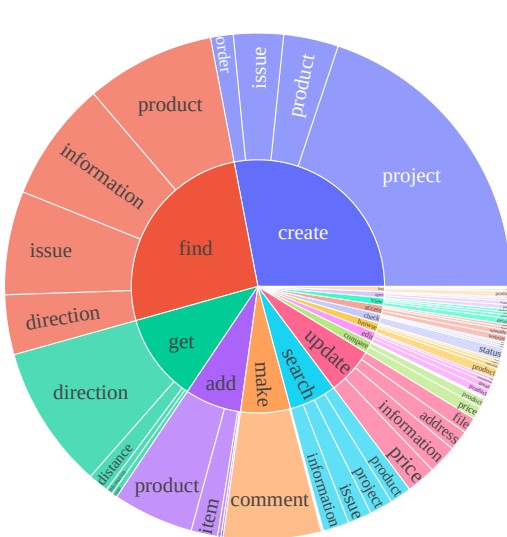
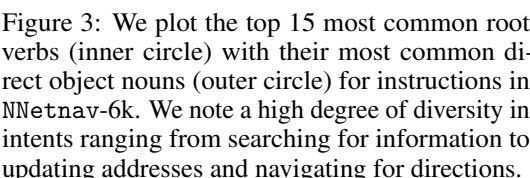

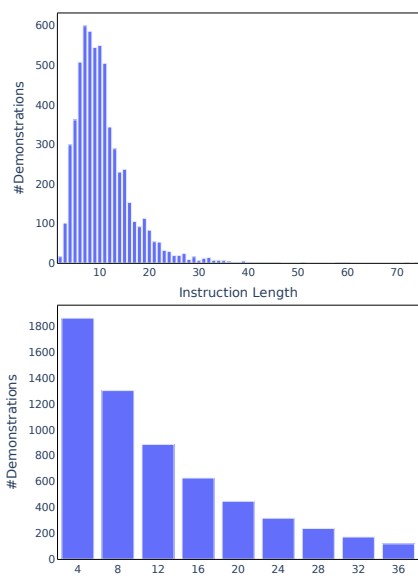

Figure 3: We plot the top 15 most common root verbs (inner circle) with their most common direct object nouns (outer circle) for instructions in `NNetnav-6k`. We note a high degree of diversity in intents ranging from searching for information to updating addresses and navigating for directions.

Figure 4: Length distribution of instructions and trajectories in `NNetnav-6k`.

**Computational savings from `NNetnav` pruning.** We visualize overall improvements in exploration efficiency in Fig 2. Each horizontal line depicts the fraction of interaction episodes that terminate at a specific time-step (indicated by the y-axis), with the red shaded area depicting additional actions that were prevented from early pruning. We find clear evidence of computational savings. In particular, over 60% of all exploration episodes were pruned after 16 actions for WebArena. For Mini-WoB++, 65% of episodes were pruned after just 4 actions in MiniWoB++, which we identify as interactions where these first actions resulted in execution failures that our pruning heuristic successfully identified.

**Self-training with `NNetnav`.** Can `NNetnav` demonstrations from an LM be used for improving the *same* LM agent? To answer this, we collect another set of `NNetnav` demonstrations on WebArena, using LM components based on `Llama-3-8B-Instruct`. Given the limitations of this smaller model, we anticipate fewer meaningful interactions. To compensate, we increase the number of episodes to 200 episodes per website, resulting in 302 demonstrations which we use for fine-tuning the same `Llama-3-8B-Instruct` agent. From results in Table 3, we find improvements of 4.3 points on WebArena.

**Cross-website generalization.** Finally, we use `NNetnav` to conduct a small study on cross-website generalization in web-agents. Concretely, we perform supervised fine-tuning of $\pi_{LM}$ on `NNetnav` demonstrations from Shopping, Maps and CMS, and evaluate generalization to Reddit and Gitlab. Here, we choose to report only model-based evaluation since success rates are 0 for these domains. From results in Table 4, we note that average reward improves by 0.06 on held out websites, and by 0.13 on in-domain websites, suggesting some potential for cross-website transfer in LLM web-agents.

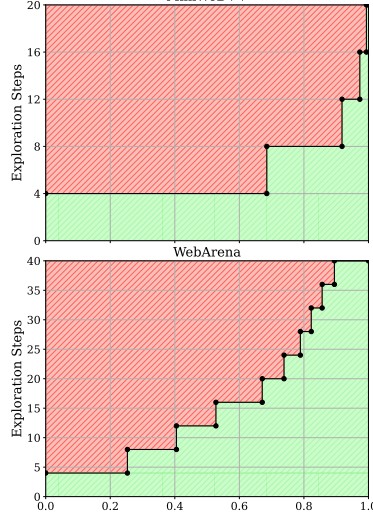

Figure 2: Horizontal lines indicate fraction of episodes terminating at corresponding y-axis exploration step. The red shaded area represents prevented actions, showing significant savings on both datasets.

| Agent | #Params | WebArena SR | Open LLM? | Zero-shot? |
|---|---|---|---|---|
| Llama-3-8B-Instruct | 8B | 1.0 | ✓ | ✓ |
| Patel et al. (2024) | 72B | 9.4 | ✓ | ✗ |
| Lai et al. (2024) | 7B | 2.5 | ✓ | ✓ |
| Ou et al. (2024) | 7B | 6.3 | ✓ | ✓ |
| Llama-3-8B-Instruct-NNetnav | 8B | **10.3** | ✓ | ✓ |
| Drouin et al. (2024) | Unknown (GPT-4) | 23.5 | ✗ | ✓ |
| Wang et al. (2024) | Unknown (GPT-4) | 35.5 | ✗ | ✗ |

Table 5: We present WebArena task success rate of various prior approaches, along with key details such as model size, the use of open LLMs, and whether methods are fully zero-shot. For Lai et al. (2024), we report results from the setting that does not use human supervision. Notably, our approach, Llama-3-8B-Instruct-NNetnav, achieves a 4% improvement over the previous state-of-the-art among zero-shot agents that use open LLMs.

## 6 NNETNAV-6K

To facilitate further research on fine-tuned browser agents, we release the first large-scale dataset of over 6000 demonstrations from WebArena. Here, we use Llama-3-70B-Instruct[3] as the underlying LM for various components in NNetnav, and sample 3000 interactions each, with $T_{max}$ set to 40 as before. For each trajectory in these demonstrations, we release both accessibility tree strings as well as browser screenshots at each time-step, to support future work on text-based as well as multi-modal web agents.

To analyze diversity in these instructions, we follow methodology from Wang et al. (2022). Specifically, we use Stanza (Qi et al., 2020) to parse each instruction, identifying the verb closest to the root and its direct object. Fig 3 presents the top 15 verbs and their corresponding object nouns. Overall, we observe a diverse range of intents in the NNetnav-6k dataset. Additionally, we plot the distribution of instruction as well as trajectory lengths in Fig 4, revealing further diversity in these aspects. Table 7 provides example demonstrations from NNetnav-6k, showcasing instructions from different websites. We find a number of complex, hierarchical instructions such as *Edit the issue "Link to WCAG 2.1 instead of ...* that refer to specific features of the website (*r/art*, *Swings and roundabouts*), and are plausible by design. Many of these instructions share lots of common structure (*e.g. Get walking directions from ...* and *Get cycling directions from ...*), and incorporating such structure into agents could be a promising direction for future work.

**Fine-tuning agents with NNetnav-6k demonstrations.** To demonstrate the effectiveness of NNetnav-6k in improving instruction-following in LLM web-agents, we perform supervised finetuning of the Llama-3-8B-Instruct agent on NNetnav-6k demonstrations. As described in Section 2, each demonstration expands into multiple training instances, resulting in a total of over 77,000 training examples. The results, presented in Table 5, show that our approach achieves a WebArena Success Rate (SR) of 10.3%. This marks a significant improvement over previously reported results for sub-10B models trained on synthetic datasets. To the best of our knowledge, our model sets a new state-of-the-art among agents that do not use closed-source models like GPT-4, human-annotated demonstrations, or prior knowledge of WebArena test instructions.

## 7 RELATED WORK

**Language Conditioned Digital Assistants.** Mapping instructions to action sequences in digital environments has been a long-standing goal in natural language understanding (Allen et al., 2007; Branavan et al., 2009). Most pre-LLM approaches for this rely on expert demonstrations for behavioral cloning (Chen & Mooney, 2011; Humphreys et al., 2022), along with an appropriately shaped reward

---

[3]We opted to use a locally hosted Llama-3-70B-Instruct model for collecting the larger-scale NNetnav-6k dataset, as it produced demonstrations of comparable quality to GPT-4o-mini while offering a more permissive license for downstream applications.

function (Branavan et al., 2009; Liu et al., 2018; Misra et al., 2017, among others). Here, learning is driven purely by synthetic demonstrations derived via (language model) exploration of websites.

**Linguistic Priors for Exploration.** Several prior works have used natural language priors to inform exploration for sequential decision making. Harrison et al. (2017) use a *trained model* of associations between language and state/action pairs to guide exploration during policy learning. Mu et al. (2022) use language annotations of states to train a goal generator module that provides intrinsic rewards for achieving generated goals. Similarly, Du et al. (2023) constrain exploration towards goals generated by a pre-trained LLM at each intermediate state of an agent. In contrast, NNetnav biases exploration through two news ways of using language priors. First, we use natural language as a way to filter meaningful interactions. Second, we use it as a pruning heuristic to navigate the potentially exponential search space of these interactions.

**Training Data for LLM Web-Agents.** LLMs have shown strong performance over a wide range of language understanding tasks, and are increasingly being used to interpret language in grounded contexts such as browsers (Yao et al., 2022; Lai et al., 2024; Wang et al., 2024; Patel et al., 2024; Lù et al., 2024, among others). Many of these approaches rely on human demonstrations, either for in-context learning (Yao et al., 2022; Sodhi et al., 2023; Kim et al., 2023) or for finetuning (Lù et al., 2024). However, because human demonstrations are costly, recent work trains LLM agents through synthetic demonstrations generated using instruction-first methods (Lai et al., 2024; Patel et al., 2024). One exception is Murty et al. (2024), which introduces an interaction-first method for generating synthetic demonstrations for in-context learning. Despite its novelty, their approach does not scale well to real websites due to the lack of a mechanism for effective exploration in environments with many possible interactions. In contrast, NNetnav also follows an interaction-first approach but improves efficiency by leveraging linguistically motivated pruning to navigate the space of meaningful interactions.

## 8  CONCLUSION

We propose NNetnav, a method for training web-agents with synthetic demonstrations. NNetnav flips the standard paradigm of synthetic data generation by first interacting with a website to generate trajectories and then relabeling trajectories into instructions. Real websites have a prohibitively large set of possible interactions; NNetnav searches over this space efficiently using a pruning function inspired by the hierarchical structure of language instructions: any complex instruction consists of language describable sub-tasks and so, if during an interaction a relabeling module cannot infer a meaningful sub-task for the trajectory-so-far, further exploration is pruned. We apply NNetnav to collect demonstrations on MiniWoB++ and WebArena, which are then used to fine-tune a zero-shot base LM agent. This yields significant improvements over the zero-shot baseline and outperforms standard synthetic data generation methods. In addition, we show that NNetnav enables self-training, as demonstrations collected using a base language model can improve the performance of an agent built on the same model. We find that NNetnav significantly enhances exploration efficiency due to the pruning heuristic and generates complex, realistic instructions. Lastly, we release NNetnav-6k, the largest dataset of demonstrations on WebArena to date, with over 6000 demonstrations covering broad range of intents and phenomena in WebArena.

## REPRODUCIBILITY STATEMENT

Prompts for every LM component is provided in Appendix A, along with other details like agent action spaces. Details for model-based evaluation on WebArena are provided in Appendix B. We provide full details for post-processing demonstrations for SFT in Appendix C, and additional hyperparameters for supervised fine-tuning in Appendix D. All code and data will be available here.

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

## A  PROMPTS FOR LM COMPONENTS

### A.1  MINIWOB++

We start by presenting all prompts for MiniWoB++. The action space for MiniWob++ is:

Listing 1: Action Space

```
noop(wait_ms: float = 1000)
    Examples:
        noop()
        noop(500)

scroll(delta_x: float, delta_y: float)
    Examples:
        scroll(0, 200)
        scroll(-50.2, -100.5)

fill(bid: str, value: str)
    Examples:
        fill('237', 'example value')
        fill('45', 'multi-line\nexample')
        fill('a12', 'example with "quotes"')

select_option(bid: str, options: str | list[str])
    Examples:
        select_option('a48', 'blue')
        select_option('c48', ['red', 'green', 'blue'])

click(bid: str, button: Literal['left', 'middle', 'right'] = 'left', modifiers: list[typing
.Literal['Alt', 'Control', 'Meta', 'Shift']] = [])
    Examples:
        click('a51')
        click('b22', button='right')
        click('48', button='middle', modifiers=['Shift'])

dblclick(bid: str, button: Literal['left', 'middle', 'right'] = 'left', modifiers: list[
typing.Literal['Alt', 'Control', 'Meta', 'Shift']] = [])
    Examples:
        dblclick('12')
        dblclick('ca42', button='right')
        dblclick('178', button='middle', modifiers=['Shift'])

hover(bid: str)
    Examples:
        hover('b8')
```

```
press(bid: str, key_comb: str)
    Examples:
        press('88', 'Backspace')
        press('a26', 'Control+a')
        press('a61', 'Meta+Shift+t')

focus(bid: str)
    Examples:
        focus('b455')

clear(bid: str)
    Examples:
        clear('996')

drag_and_drop(from_bid: str, to_bid: str)
    Examples:
        drag_and_drop('56', '498')

upload_file(bid: str, file: str | list[str])
    Examples:
        upload_file('572', 'my_receipt.pdf')
        upload_file('63', ['/home/bob/Documents/image.jpg', '/home/bob/Documents/file.zip
        '])

Only a single action can be provided at once. Example:
fill('a12', 'example with "quotes"')

If you are done exploring, you can issue the stop action: '''stop'''

Here is an example with chain of thought of a valid action when clicking on a button: "In
order to accomplish my goal I need to click on the button with bid 12. In summary, the next
 action I will perform is '''click("12")'''
```

This is then directly used for various prompts as {action_str}.

### Listing 2: Prompt for the Exploration Policy $\pi_{explore}$

```
You are an autonomous intelligent agent tasked with performing tasks on a web interface.
Your objective is to simulate a task that a person might request, by interacting with the
interface through the use of specific actions.

Here's the information you'll have:
DOM Representation: This is the current webpage's Document Object Model (DOM)
representation as a string.
The previous action: This is the action you just performed. It may be helpful to track your
 progress.
Trajectory: This is a sequence of natural language descriptions of the agent's interaction
with the web-browser.
Person Description: The description of a specific kind of person whose task you are
supposed to simulate.

You can perform the following actions: {action_str}

To be successful, it is very important to follow the following rules:
1. You should only issue an action that is valid given the current observation.
2. You should only issue one action at a time.
3. You should reason step by step and then issue the next action.
4. Make sure to wrap your action in a code block using triple backticks.
5. The DOM / Accessibility Tree only shows the visible part of the webpage. If you need to
interact with elements that are not visible, you can scroll to them using the scroll action
. Often submit buttons are not visible and are at the bottom of the page. To scroll to the
bottom of the page, use the scroll action with a large value for the y coordinate.
6. To generate an interesting task, make sure you issue atleast 4 actions before stopping.
More interesting tasks typically involve more interactions with the browser.
7. You can issue atmost 20 actions before stopping, but feel free to output the stop action
 early if you want to stop exploring. Don't generate anything after stop.
```

### Listing 3: Prompt for $\Delta_{LM}$

```
You are given the output of an action taken by an autonomous intelligent agent navigating a
 web-interface to fulfill a task given by a user.  Your objective is to produce a
description of the changes made to the state of the browser.
```

```
Here's the information you'll have:
Initial state of the browser as a DOM representation: This is the webpage's Document Object
 Model (DOM) representation as a string.
Final state of the browser as a DOM representation: This is the DOM representation after
the agent took the action.

The action taken by the agent: This is the action taken by the agent to change the state of
 the browser.

The actions the agent can take come from the following categories: {action_str}

To be successful, it is very important to follow the following rules:
1. Explictly think about the various features on the website and how the interaction with
the website changed these features
2. Provide the description of changes in one or two sentences.
3. Strictly follow the format "State change: <your-answer>" for your output
```

Listing 4: Prompt for the Trajectory Labeling function $Lf_{LM}$

```
Given a task from a user, an autonomous intelligent agent carries out a sequence of actions
 on a web-interface.

The actions the agent can take fall under the following categories: {action_str}

Your objective is to guess the instruction the user gave, given the following information:
Trajectory: This is a sequence of natural language descriptions of the agent's interaction
with the web-browser.

To be successful, it is very important to follow the following rules:
1. Explictly think about how the trajectory is a valid way to achieve the instruction,
before outputing the instruction.
2. Start by thinking by outputing Thought: <your-reasoning>.
3. End your answer by strictly following the format "Instruction: <your-answer>" for your
output.
```

Listing 5: Prompt for the reward function $s_{LM}$

```
An autonomous intelligent agent navigating a web browser is given an instruction by a user.
 Your objective is to give a score to the agent based on how well it completed its task.
Your score must be on the scale of 1 to 5. Give a score of 5 only when there are no errors.
 To do this task you are provided with the following information:

Instruction: This is the natural language instruction given to the agent.
Trajectory: This is a sequence of natural language descriptions of the agent's interaction
with the web-browser.

To be successful, it is very important to follow the following rules:
1. Explictly think about what is needed to follow the instruction correctly on the website
and if the trajectory reflects these steps.
2 Give a score of 4 if there are very minor errors, or if the task was more than 70%
completed. Give a score of 3 (or below) if the model made very little progress towards the
given instruction or if there are major errors.
3. Start by thinking by outputing Thought: <your-reasoning>.
4. End your answer by strictly following the format "Reward: <your-answer>" for your output
```

Listing 6: Prompt for the base LLM agent $\pi_{LM}$

```
You are an autonomous intelligent agent tasked with performing tasks on a web interface.
These tasks will be accomplished through the use of specific actions you can issue.

Here's the information you'll have:
DOM Representation: This is the current webpage's Document Object Model (DOM)
representation as a string.
The user's objective: This is the task you're trying to complete.
The previous action: This is the action you just performed. It may be helpful to track your
 progress.

You can perform the following actions: {action_str}

To be successful, it is very important to follow the following rules:
```

```
1. You should only issue an action that is valid given the current observation
2. You should only issue one action at a time.
3. You should follow the examples to reason step by step and then issue the next action.
4. Make sure to wrap your action in a code block using triple backticks.
5. The DOM / Accessibility Tree only shows the visible part of the webpage. If you need to
interact with elements that are not visible, you can scroll to them using the scroll action
. Often submit buttons are not visible and are at the bottom of the page. To scroll to the
bottom of the page, use the scroll action with a large value for the y coordinate.
6. Issue stop action when you think you have achieved the objective. Don't generate
anything after stop.
```

Listing 7: Prompt for adding reasoning steps retroactively to filtered trajectories

```
You are an autonomous intelligent agent that carries out a sequence of actions on a web-
interface, given an instruction from a user. The actions you can take fall under the
following categories: {action_str}

You are given the user instruction, an intermediate state of the web-page (in the form of a
 DOM string), and the action you took for that intermediate state. Your objective is to
output your reasoning for choosing that specific action. In summary, you are given the
following
Instruction: This is the instruction given by the user.
Intermediate State: This is the state of the web-page at some time-step t.
Action: This is the action taken by the agent at time-step t.

Here are some example reasoning outputs for some random tasks:

Instruction: select_option('checkbox_group', ['appetizing', 'bunnies', 'weird', 'dull'])
Intermediate State: {state}
Action: click('21')

Output: Let's think step by step. The user instruction is to select multiple options from a
 checkbox group, specifically 'appetizing', 'bunnies', 'weird', and 'dull'. In the current
intermediate state, I can see the checkboxes for these options, and they are represented by
 input elements with specific bid attributes. The action I took was to click on the
checkbox corresponding to 'appetizing', which is represented by the input with bid '18'.
Now I will check the 'bunnies' checkbox, moving closer to fulfilling the user's request to
select the specified options. In summary, the next action I will perform is ```click('21')
```

Instruction: click('Macie')
Intermediate State: {state}
Action: click('21')

Output: Let's think step by step. The user instruction is to click on 'Macie', which is
represented by the HTML element with bid '21'. In the intermediate state of the web page, '
Macie' is identified as a folder that is expandable and clickable. By executing the click
action on this element, I will open the folder associated with 'Macie', which aligns with
the user's intention to interact with it. In summary, the next action I will perform is ```
click('21')```

To be successful, it is very important to follow the following rules:
1. Explictly think about how executing the given action will change the web-page in a way
that gets the agent closer to achieving the user instruction
2. You should strictly follow the format "Output: Let's think step by step. <agent
reasoning>. In summary, the next action I will perform is ```[action]``` ". Here make sure
to replace [action] with the provided action.
```

## A.2 WEBARENA

Next, we present all prompts for WebArena. The action space for WebArena is:

Listing 8: Action Space

```
Page Operation Actions:
`click [id]`: This action clicks on an element with a specific id on the webpage.
`type [id] [content] [press_enter_after=0|1]`: Use this to type the content into the field
with id. By default, the "Enter" key is pressed after typing unless press_enter_after is
set to 0.
`hover [id]`: Hover over an element with id.
`press [key_comb]`:  Simulates the pressing of a key combination on the keyboard (e.g.,
Ctrl+v).
```

```
'scroll [direction=down|up]': Scroll the page up or down.

Tab Management Actions:
'new_tab': Open a new, empty browser tab.
'tab_focus [tab_index]': Switch the browser's focus to a specific tab using its index.
'close_tab': Close the currently active tab.

URL Navigation Actions:
'goto [url]': Navigate to a specific URL.
'go_back': Navigate to the previously viewed page.
'go_forward': Navigate to the next page (if a previous 'go_back' action was performed).

Completion Action:
'stop ["done"]': Issue this action when you are done.

Homepage:
If you want to visit other websites, check out the homepage at http://homepage.com. It has
a list of websites you can visit.
```

This is then directly used for various prompts as `{action_str}`.

Listing 9: Prompt for the Exploration Policy $\pi_{\text{explore}}$

```
You are an autonomous intelligent agent tasked with navigating a web browser.  Your
objective is to simulate a task that a person might perform, by interacting with the
browser through the use of specific actions.

Here's the information you'll have:

The current web page's accessibility tree: This is a simplified representation of the
webpage, providing key information.
The current web page's URL: This is the page you're currently navigating.
The open tabs: These are the tabs you have open.
The previous action: This is the action you just performed. It may be helpful to track your
 progress.
Trajectory: This is a sequence of natural language descriptions of the agent's interaction
with the web-browser.
Person Description: The description of a specific kind of person whose task you are
supposed to simulate.

The actions you can perform fall into several categories: {action_str}

To be successful, it is very important to follow the following rules:
1. You should only issue an action that is valid given the current observation
2. You should only issue one action at a time.
3. You should follow the examples to reason step by step and then issue the next action.
4. Generate the action in the correct format. Start by reasoning out the current situation.
 End with "In summary, the next action I will perform is" phrase, followed by action inside
 ``````. For example, "Let's think step-by-step. Given the current state, I need to click
on the like button which has id 1234. In summary, the next action I will perform is ```
click [1234]```".
5. To generate an interesting task, make sure you issue atleast 4 actions before stopping.
More interesting tasks typically involve more interactions with the browser.
6. You can issue atmost 40 actions before stopping, but feel free to output the stop action
 early if you want to stop exploring. Don't generate anything after stop.

Here are some example outputs for some random tasks:
1. Let's think step-by-step. This page list the information of HP Inkjet Fax Machine, which
 is the product identified in the objective. Its price is $279.49. I think I have achieved
the objective. I will issue the stop action with the answer. In summary, the next action I
will perform is ```stop [$279.49]```
2. Let's think step-by-step. This page has a search box whose ID is [164]. According to the
 nominatim rule of openstreetmap, I can search for the restaurants near a location by "
restaurants near". I can submit my typing by pressing the Enter afterwards. In summary, the
 next action I will perform is ```type [164] [restaurants near CMU] [1]``
```

Listing 10: Prompt for $\Delta_{\text{LM}}$

```
You are given the output of an action taken by an autonomous intelligent agent navigating a
 web browser.  Your objective is to produce a description of the changes made to the state
of the browser.
```

```
Here's the information you'll have:

Initial state of the browser as an accessibility tree: This is a simplified representation
of the webpage, providing key information.
Final state of the browser: This is the accessibility tree representation after the agent
took the action

The action taken by the web agent: The agent can take actions that fall under the following
 categories: {action_str}

To be successful, it is very important to follow the following rules:
1. Explictly think about the various features on the website and how the interaction with
the website changed these features
2. Provide the description of changes in one or two sentences.
3. Strictly follow the format "State change: <your-answer>" for your output
```

Listing 11: Prompt for the Trajectory Labeling function $Lf_{LM}$

```
Given an instruction from a user, an autonomous intelligent agent carries out a sequence of
 actions on a web-browser. The actions the agent can take fall under the following
categories: {action_str}

Your objective is to guess the instruction the user gave, given the following information:
Trajectory: This is a sequence of natural language descriptions of the agent's interaction
with the web-browser.

Here are some examples of user instructions:
1. Get the distance from SF airport to Palo Alto.
2. Find out the price of Apple airpods
3. Add 5 items to cart
4. Make a comment on the first post in the r/gaming subreddit.

To be successful, it is very important to follow the following rules:
1. Explictly think about how the trajectory is a valid way to achieve the instruction,
before outputing the instruction.
2. Start by thinking by outputing Thought: <your-reasoning>.
3. End your answer by strictly following the format "Instruction: <your-answer>" for your
output.
```

Listing 12: Prompt for the reward function $s_{LM}$

```
An autonomous intelligent agent navigating a web browser is given an instruction by a user.
 Your objective is to give a score to the agent based on how well it completed its task.
Your score must be on the scale of 1 to 5. Give a score of 5 only when there are no errors.
 To do this task you are provided with the following information:

Instruction: This is the natural language instruction given to the agent.
Trajectory: This is a sequence of natural language descriptions of the agent's interaction
with the web-browser.

To be successful, it is very important to follow the following rules:
1. Explictly think about what is needed to follow the instruction correctly on the website
and if the trajectory reflects these steps.
2 Give a score of 4 if there are minor errors, or if the task was more than 70% completed.
Give a score of 3 (or below) if the model made very little progress towards the given
instruction.
3. Start by thinking by outputing Thought: <your-reasoning>.
4. End your answer by strictly following the format "Reward: <your-answer>" for your output
```

Listing 13: Prompt for the base LLM agent $\pi_{LM}$

```
You are an autonomous intelligent agent tasked with navigating a web browser. You will be
given web-based tasks. These tasks will be accomplished through the use of specific actions
 you can issue.

Here's the information you'll have:
The user's objective: This is the task you're trying to complete.
The current web page's accessibility tree: This is a simplified representation of the
webpage, providing key information.
The current web page's URL: This is the page you're currently navigating.
The open tabs: These are the tabs you have open.
```

```
The previous actions: These are all the action you have performed. It may be helpful to
track your progress.

The actions you can perform fall into several categories: {action_str}

To be successful, it is very important to follow the following rules:
1. You should only issue an action that is valid given the current observation
2. You should only issue one action at a time.
3. You should follow the examples to reason step by step and then issue the next action.
4. You are strictly forbidden from issuing a goto action to a URL that is not on the
homepage.
5. Generate the action in the correct format. Start by reasoning about the current
situation. End with "In summary, the next action I will perform is" phrase, followed by
action inside ``````. For example, "Let's think step-by-step. Given the current state, I
need to click on the like button which has id 1234. In summary, the next action I will
perform is ```click [1234]```".
6. Issue stop action when you think you have achieved the objective. Don't generate
anything after stop.

Here are some example outputs for some random tasks:
1. Let's think step-by-step. This page list the information of HP Inkjet Fax Machine, which
 is the product identified in the objective. Its price is $279.49. I think I have achieved
the objective. I will issue the stop action with the answer. In summary, the next action I
will perform is ```stop [$279.49]```
2. Let's think step-by-step. This page has a search box whose ID is [164]. According to the
 nominatim rule of openstreetmap, I can search for the restaurants near a location by "
restaurants near". I can submit my typing by pressing the Enter afterwards. In summary, the
 next action I will perform is ```type [164] [restaurants near CMU] [1]```
```

Listing 14: Prompt for adding reasoning steps retroactively to filtered trajectories

```
You are an autonomous intelligent agent that carries out a sequence of actions on a web-
interface, given an instruction from a user.

The actions you can take fall under the following categories: {action_str}

You are given the user instruction, an intermediate state of the web-page (in the form of
an accessibility tree), and the action you took for that intermediate state. Your objective
 is to output your reasoning for choosing that specific action. In summary, you are given
the following
Instruction: This is the instruction given by the user.
Intermediate State: This is the state of the web-page at some time-step t.
Action: This is the action taken by the agent at time-step t.

Here are some example reasoning outputs for some random tasks

Instruction: Find directions from CMU to Downtown Pittsburgh
Intermediate State: {state}
Action: click [482]

Output: Let's think step-by-step. Since my previous attempt to click the 'Go' button failed
, I will try clicking the 'Find directions between two points' link instead. This could
help in planning public transportation routes effectively. In summary, the next action I
will perform is ```click [482]```

Instruction: Navigate to the 'Orders' section and create a new customer for order
processing.
Intermediate State: {state}
Action: click [1583]

Output: Let's think step-by-step. Currently, I am in the Orders section of the Magento
Admin panel. I see a button labeled 'Create New Order,' which is likely the next step for
creating orders. In summary, the next action I will perform is ```click [1583]``` to create
 a new order.

To be successful, it is very important to follow the following rules:
1. Explictly think about how executing the given action will change the web-page in a way
that gets the agent closer to achieving the user instruction
2. Make sure to wrap the action inside triple backticks (such as ```click [1234]```, ```
type [12] [Hotels near CMU]```) as shown in the examples and strictly follow the format "
Output: Let's think step by step. <agent reasoning>. In summary, the next action I will
perform is ```[action]``` ". Here make sure to replace [action] with the provided action.
```

Listing 15: Prompt for appending the special [stop] action in WebArena

```
Given an instruction from a user, an autonomous intelligent agent carries out a sequence of
 actions on a web-browser. The actions the agent can take fall under the following
categories (we also provide the descriptions of each action): {action_str}

You are given the user instruction, and the final webpage after the agent finished its task
. Unfortunately, we forgot to collect the final stop action from the agent. Your objective
is to guess the agent's stop action. To do this, you are given the following
Instruction: This is the instruction given by the user.
Final State: This is the final state of the web-page after the agent executed its actions
on the browser.

Here are some examples of valid outputs:
1. Let's think step-by-step. The task requires me to find the person with the most number
of upvotes. I see the answer to that is Alice Oh. Therefore I will stop now. In summary, my
 next action will be '''stop [Alice Oh]'''.
2. Let's think step-by-step. The task required setting the price of Sprite to 25$ which I
have already done. Thus I will stop now. In summary, my next action will be '''stop [N/A
]'''.
3. Let's think step-by-step. I was supposed to find the distance from Brad's house to the
coffee shop. I see this info on the map as 0.3 miles. Thus I will issue the stop action. In
 summary, my next action will be '''stop [0.3 miles]'''

To be successful, it is very important to follow the following rules:
1. Explictly think about what kind of a stop action was needed. For instance, if the user
requests information (e.g. Search for airports near CMU or Find developers with more than 5
 merge requests), then the stop action should have the answer based on the final web-page (
e.g. '''stop [Pittsburgh Airport]''' or '''stop [Don Knuth, Alan Turing]'''). Otherwise,
the stop action should be without any arguments (e.g. '''stop''').
2. Your output should include reasoning steps. Also make sure to wrap the stop action in
triple backticks for e.g. '''stop [<your answer>]'''. Overall, follow the following format
for your output: "Let's think step by step. <your reasoning>. In summary, my next action
should be '''stop [<your answer>]'''.
```

## B  MODEL-BASED EVALUATION: DETAILS

For each $(g, \tau)$ pair we first use $\Delta_{\text{LM}}$ to compute the sequence of changes $\delta_\tau$, which is then passed into the reward module along with $g$. We implement the reward module as a prompted LM, using the largest GPT-4o (specifically `gpt-4o-2024-08-06`) with the following prompt:

Listing 16: Prompt for the model-based evaluator

```
An autonomous intelligent agent navigating a web browser is given an instruction by a user.
 Your objective is to give a score to the agent based on how well it completed its task.
Your score must be on the scale of 1 to 5. Give a score of 5 only when there are no errors.
 To do this task you are provided with the following information:

Instruction: Natural language instruction given to the agent.
Trajectory: Sequence of language descriptions of the agent's interaction with the browser.

Here are some guidelines for scoring:
1. Give a score of 5 if there are no errors.
2. Give a score of 4 if the task was almost correctly done (i.e. for form filling, most of
the fields are filled or for a search task, a query was correctly typed, and the agent
navigated to the right links).
3. Give a score of 3 if the task was only partially completed (i.e for form filling, less
than half the fields are filled out) and if there are other minor execution errors.
4. Give a score of 1 or 2 if there are major execution errors, or the task was hardly
completed, or if the agent did something completely unrelated.

To be successful, it is very important to follow the following rules:
1. Explictly think about what is needed to follow the instruction correctly on the website
and if the trajectory reflects these steps.
2. Start by thinking by outputing Thought: <your-reasoning>.
3. End your answer by strictly following the format "Reward: <your-answer>" for your output
```

## C  PROCESSING DEMONSTRATIONS FOR SFT

As mentioned in §2, for supervised finetuning each demonstration is converted into multiple training instances. We perform this conversion differently based on input features of $\pi_{\text{LM}}$.

| Dataset | NNetnav | NNetnav (self-train) | Instruction-First |
|---------|---------|----------------------|-------------------|
| MiniWoB++ | 2288 | - | 8559 |
| WebArena | 9737 | 2204 | 1681 |

Table 6: Number of instances for supervised training experiments of §5 under various settings. Between NNetnav and Instruction-First, we only control for the number of episodes for a fair comparison, which results in different number of training instances.

**MiniWoB++.** For MiniWoB++, $\pi_{\text{LM}}$ conditions on the current observation $o_t$, the goal $g$ and the previous action $a_{t-1}$ (see prompt in §A.1). Thus, we pre-process each $(g, \tau)$ demonstration into inputs $(g, o_t, a_{t-1})$ with the corresponding reasoning step and action $(r_t, a_t)$ as the target output.

**WebArena.** For WebArena, $\pi_{\text{LM}}$ conditions on the current observation $o_t$, the goal $g$ and all previous actions $\{a_1, a_2, \ldots, a_{t-1}\}$ (see prompt in §A.2). Thus, we pre-process each $(g, \tau)$ demonstration into inputs $(g, o_t, \{a_{<t}\})$ with $(r_t, a_t)$ as the target output.

We report number of training instances from NNetnav and instruction-first generation for both environments in Table 6.

## D  TRAINING DETAILS

**Additional Hyperparameters.** For all Llama-3-8B-Instruct finetuning experiments, we set the batch size for training as $128 \times 4096$, train for 5 epochs, with a learning rate of 2e-5 that linearly warms up from 0 over 3% of total training steps. We use 4 A100 GPUs with 80GB GPU memory, and additionally use DeepSpeed ZeRO-3 (Rajbhandari et al., 2020) to speed up training and manage memory.

## E  NNETNAV-6K EXAMPLES

| Shopping |
|---|
| Find a kitchen utensil organizer. |
| Find a kitchen utensil organizer within a certain budget. |
| Write a review for the product "Citric Acid 2 Pounds 100% Pure Organic Food Grade". |
| Find the price of kitchen gadgets that can be used for dining and entertaining, and add them to the cart. |
| Browse for women's clothing items, specifically jumpsuits, and add some to cart. |
| **CMS** |
| Change the stock status of the Sprite Stasis Ball 65 cm to In Stock. |
| Create a new product in the Magento Admin panel with the name 'New Fashionable Watch', SKU 'New Fashionable WatchFW101', price $100.00, and set as new from 2024-01-01. |
| Update the price of Sprite Stasis Ball 55 cm to $24.50 and set its quantity to 50. |
| Add two products, "Abominable Hoodie" and "Samsung Smart TV", with respective prices $99.99 and $50.00, and then start the process of adding a new customer. |
| **Reddit** |
| Create a new forum called "Funny Stuff" with the title "Memes and LOLs", description "A place for sharing and discussing funny memes and LOLs", and sidebar "Memes of the day". |
| Find a webpage related to intraday trading strategies on the wallstreetbets forum. |
| Find and participate in a discussion on the wallstreetbets forum about intraday trading strategy, specifically on a post titled "Swings and roundabouts". |
| Change my profile settings to use Deutsch as the language and Africa/Accra as the time zone, and then view the search results for "r/art". |
| **Maps** |
| Get walking directions from Logan Street, Pittsburgh, PA to Carnegie Mellon University on OpenStreetMap. |
| Get the cycling directions from Brooklyn to Manhattan. |
| Find the driving directions from TLC Medical Transportation Services in Syracuse to Times Square in Manhattan. |
| **Gitlab** |
| Create a new project named 'My Blog Post Project' and add an Apache License 2.0 file. |
| Create a new project, add a LICENSE file with Apache License 2.0, and approve the "Add verification functions" merge request. |
| Search for a README.md file within the "My New Project" project and verify its contents. |
| Edit the issue "Link to WCAG 2.1 instead of 2.0?" in the First Contributions project on GitLab by updating its title and description to point to WCAG 2.1 guidelines instead of 2.0 guidelines. |
| Investigate the node-http-proxy project's issue #992 regarding connection headers and determine its relevance to the Byte Blaze project. |
| Investigate and comment on the "Outdated dependencies" issue in the "Byte BlazeByte BlazeByte Blaze / accessible-html-content-patterns" project. |

Table 7: Some Example demonstrations obtained from `NNetnav-6k`. We note that these instructions are hierarchical, refer to concrete features and entities and plausible by design.

