# OpenReview forum: "NNetscape Navigator: Complex Demonstrations for Web Agents Without a Demonstrator"
_ICLR.cc/2025/Conference — ICLR 2025 Conference Withdrawn Submission_

### Official Review · Reviewer_Ezws · 2024-10-28

**Soundness:** 2
**Presentation:** 2
**Contribution:** 2
**Rating:** 5
**Confidence:** 4

**Summary:**

The authors propose a method for synthetic data collection of web trajectories for llm finetuning. The method does the following:
- Use an LLM to explore the web based on a predefined persona.
- Prompt the LLM to predict state changes between consecutive actions.
- From the sequence of state changes, generate a plausible instruction for the trajectory using the LLM.
- Evaluate instruction quality with an LLM-based reward function.
- Finetune an LLM on the generated and labeled trajectories.

**Strengths:**

- Proposes a method for generating synthetic trajectories using LLM exploration and hindsight relabeling for web applications.
- Demonstrates that training on these trajectories leads to better performance compared to instruction-first trajectories and zero shot baselines on web arena.
- Synthetic data generation without human annotation for improving llm agents on the web is an important problem, and well motivated in the paper.

**Weaknesses:**

- Lack of clarity on whether the reward LLM is applied to filter the instruction-first baseline.
- Insufficient details on the use of the reward model and filtering of incorrect trajectories.
- No error analysis or discussion of limitations.
- Performance gains are minor for certain websites.
- No comparison with previous methods that report results on the web arena.

**Questions:**

1. How exactly is the reward score used? Why is the reward function only applied when R=0? How do you determine the threshold for the 1-5 rating? Why stop data collection at R=0 instead of discarding the sample and continuing the trajectory?
2. Is there any filtering applied to the instruction-first SFT baseline, such as using the reward LLM to filter bad trajectories?
3. Why use different LLMs for NNetNav collection (GPT) and training (LLaMA)? Can the same LLM be used for both collection and inference?
4. Why is there no improvement on certain tasks, like Reddit and GitLab? More analysis of the errors and limitations in the collection method is needed.

---

> ### Author Response · Authors · 2024-11-18
> **Response + new results**
>
> We thank Ezws for their thoughtful feedback. We provide clarifications below:
>
> Clarity on the reward model for the instruction-first baseline:  Yes, we confirm that we use the same reward model described in Lines 199-200 for both the instruction-first baseline and NNetNav. While this is briefly mentioned in Lines 138-146, we apologize for any lack of clarity and will revise the draft to make this more explicit.
>
> Using the reward model for filtering: Please see our response to Q1 below for a detailed response on how the reward model is used.
>
> Performance gains minor for certain websites: We attribute this to limitations in the WebArena success rate metric. When we utilize a model-based evaluator, we observe more consistent performance improvements across all websites as can be seen in Table-2. Notably, such model-based evaluators are becoming more common in the community (Pan et al. 2024, Zhuge et al. 2024).
>
> ### Comparison with  previous methods that report results on WebArena:
>
> Thank you for the suggestion. We are happy to provide comparisons with prior results on WebArena using the official leaderboard [https://shorturl.at/TFf5q]. Except for AutoWebGLM, which heavily leverages human demonstrations, all methods in the sub-10B category achieve below a 7% success rate. In contrast, our model achieves a 7.2% success rate, establishing a new state-of-the-art at this scale without relying on human demonstrations. Additionally, following feedback from other reviewers, we conducted a new experiment where we fine-tuned LLaMA-3-8b on NNetNav-6k. As detailed in Section 2, the 6,000 demonstrations are expanded into over 77,000 training instances due to multiple training instances per trajectory. Using the same hyperparameters as used in the paper, fine-tuning LLaMA-8b-instruct resulted in a WebArena success rate of 10.3%—a significant increase over the previously reported 7.2% (where we used a smaller dataset for fair comparisons with other baselines). **This result surpasses previously published results for sub-10B models using synthetic datasets and, to our knowledge, establishes our model as the state-of-the-art among agents that do not use closed-source models (like GPT-4 or GPT-3.5), or human demonstrations**:
>
> * LLaMA-8b-NNetNav [ours]: 10.3% (6k demonstrations)
> * AutoWebGLM-7b (S1): 2.5% with 240 demonstrations
> * Synatra-CodeLLaMA-7b: 6.3% with 30k demonstrations
> * Patel et al. (2024): 9.36% with Qwen-1.5-72B. Our results surpass those reported by Patel et al. (2024), a prior work mentioned by nEkA, who used a significantly larger Qwen-1.5-72B-Chat model.
>
> These results highlight that our approach is the current state-of-the-art among methods relying solely on synthetic demonstrations, without utilizing closed-source models. We hope these updated comparisons offer a clearer perspective on the advancements made by our work.
>
> ---
> Next, we answer questions from Ezws below:
>
> Using the reward score: We appreciate the inquiry regarding the reward score. To clarify further, our reward model uses a prompted LLM that takes the trajectory summary and the instruction as input, and outputs a score between 1 and 5 (see the prompt in Listing 5 of the appendix). If the score is less than 4, the output is set to 0; otherwise, it is set to 1. This determines whether we should prune or continue exploring, as outlined in Algorithm-1. Notably, the threshold is not manually set; rather, we rely on the language model to generate an appropriate rating. While this method may introduce some noise, it performs well in practice.
>
> Filtering for Instruction-first SFT:  Yes, the same reward model is used for filtering demonstrations in the Instruction-first sampling. Here, a demonstration is kept only if the reward score is 1; otherwise, it is discarded. We will clarify this more explicitly in the manuscript (though it is briefly mentioned in Lines 138-145).
>
> Using different LMs for NNetNav and training:  Indeed, the same LLM can be used for data collection, training, and inference. *We have already conducted this experiment and report the results in Table-3 (see Lines 405-411), using LLaMA-3-8b for both data collection and inference*. Here, we find improvements of over 4 points in webarena success rate, entirely from self-training.
>
> Improvements on Reddit and Gitlab: We do observe improvements on Reddit and Gitlab (refer to Table-2). However, due to inherent limitations in the WebArena success rate metric—which hard-codes certain solutions and does not provide partial credit—these improvements may not be fully reflected. For instance, even if a model's performance on a task improves from 0% to nearly 90%, the success rate remains 0% under the current metric. A graded evaluation, however, would capture these gains.
>
> We hope these clarifications adequately address your concerns, and we are committed to revising the manuscript to enhance clarity based on your valuable feedback. Thank you once again for your insightful comments.

---

> ### Comment · Reviewer_Ezws · 2024-11-25
> **Thank you**
>
> I thank the authors for their thoughtful responses and for clarifying the instruction-first baseline and the use of different LMs. The proposed approach of relabeling and rejection sampling is interesting, and I acknowledge the performance of the sub-10B models using synthetic datasets. However, I remain concerned about the method's limitations, given the lack of discussion and analysis of failure modes and scalability compared to other approaches. As such, I will maintain my rating marginally below the acceptance threshold.

---

### Official Review · Reviewer_nEkA · 2024-11-02

**Soundness:** 2
**Presentation:** 2
**Contribution:** 2
**Rating:** 3
**Confidence:** 5

**Summary:**

This paper proposes NNetscape Navigator (NNetnav), a pipelined approach where firstly exploration LLMs (gpt-4o-mini, Llamma-3-8B-Instruct, Llamma-3-70B-Instruct, etc) generate the random exploration trajectory without specific instructions, and then provide “hindsight” instructions to the collected exploration trajectories while removing redundant actions. By leveraging such synthetic demonstrations for finetuning, LLM agents (based on Llamma-3-8B-Instruct) can improve their performance. The experiments are conducted on MiniWoB++ and WebArena. The paper is planning to release the synthesized trajectories, NNetnav-6K, to the community.

**Strengths:**

- [S1] Compared to the baseline methods (Zero-Shot, SFT(instruction-first), SFT(NNetnav+Distil.)), NNetnav successfully improve the preformance on MimiWoB++ and WebArena.

**Weaknesses:**

- [W1] The proposed pipeline is quite similar to the BAGEL (Murty et al., 2024), which proposes an exploration and relabeling approach for data collection in web navigation. BAGEL proposes both instruction-first, and demonstration-first methods, and demonstrates that the demonstration-first method performs better than the instruction-first one, which is the same consequence as this paper. In this sense, I think the novelty is very limited.

- [W2] It is unclear whether the achieved performance is meaningful to the community or not. For instance, MiniWoB has been well-studied and is the benchmark where capable LLM-based agents can resolve almost perfectly (90-95%; e.g. Kim et al., 2023, Zheng et al., 2023, Furuta et al., 2023). It is good to see the improvement from the baselines, but achieving 48% success does not provide progress in this area. WebArena also has 10-15%-success-rate baselines and over 30% success SoTA agents. The 7.2% success is actually not a good result. In addition, Patel et al., (2024) also propose a synthetic data collection pipeline in web agents, which starts with  7.14% success and results in 9.36% success. The methodologies of those two are similar, and the absolute performance of this paper does not provide progress on the benchmark.

- [W3] The effectiveness/quality of NNetNav-6K is quite unclear. As far as I read the paper, this dataset is different from the data where the agents were trained in this paper (for table 1 or 3; collected by gpt-4o-mini or Llamma-3-8B-Instruct). I think there are no experiments with NNetNav-6K. Also, the dataset size for the main experiment is 9.7K (from Table 6), but the future-released data is smaller (6K). Considering the performance improvement is marginal in the main experiments, it is very hard to imagine that NNetNav-6K can provide a significant gain.

Considering those aspects, I vote for the rejection.

**Reference**

- Murty et al., 2024. BAGEL: Bootstrapping Agents by Guiding Exploration with Language. https://arxiv.org/abs/2403.08140
- Patel et al., 2024. Large Language Models Can Self-Improve At Web Agent Tasks https://arxiv.org/abs/2405.20309
- Kim et al., 2023. Language Models can Solve Computer Tasks https://arxiv.org/abs/2303.17491
- Zheng et al., 2023. Synapse: Trajectory-as-Exemplar Prompting with Memory for Computer Control https://arxiv.org/abs/2306.07863
- Furuta et al., 2023. Multimodal Web Navigation with Instruction-Finetuned Foundation Models https://arxiv.org/abs/2305.11854

**Questions:**

See the weakness above.

---

> ### Author Response · Authors · 2024-11-18
> **Thanks for your review + new results establishing SoTA among open-source sub 10B agents**
>
> We thank nEkA for their review and feedback. We address points raised by nEkA below:
>
> ### Differences with BAGEL
>
> We appreciate the opportunity to clarify the differences between BAGEL and our work. We would like to draw nEkA’s attention to Lines 510-515 of the manuscript, where we discuss these differences. Additionally, as mentioned in the abstract (Lines 015-017), prior interaction-first techniques, such as BAGEL, do not address efficient exploration in complex environments. For convenience, we provide a summary of the differences below:
> * Algorithmic Differences: BAGEL employs a flat exploration strategy effective for simpler environments like MiniWoB++ but becomes inefficient on complex, real-world websites. In contrast, NNetNav introduces a novel exploration method that leverages inferred sub-tasks to prune exploration paths early, significantly improving efficiency. As demonstrated in Figure 2, this results in nearly halving the number of interactions with the environment.
> * Experimental Differences: While BAGEL is limited to simpler environments, our experiments extend NNetNav’s application to complex, real-world websites. Additionally, we fine-tune large-scale models (8 billion parameters) for our experiments, whereas BAGEL uses a RAG-based approach. Further, we are releasing NNetNav6k, the largest dataset to date for demonstrations on WebArena. In contrast, BAGEL has not released a dataset.
> While both approaches share a high-level motivation centered around interaction-first data collection, the specific methodologies and experimental setups are notably distinct. We hope this strengthens nEkA's confidence in the novelty of our approach.
>
>
> ### Performance compared to prior work
>
> * We would like to clarify that the methods by Kim et al. and Furuta et al. are not unsupervised for MiniWoB++. Specifically, Kim et al. utilize domain-specific prompts, while Furuta et al. leverage the reward function. Similarly, Zheng et al. rely on human demonstrations for in-context learning. More importantly, these works use significantly larger closed-source models, such as GPT-3.5 and GPT-4, whereas we employ a much smaller, open-source LLaMA-8b model and achieve notable performance improvements of 20 points through fine-tuning with NNetNav demonstrations.
> * Regarding WebArena, we respectfully note that prior works mentioned by nEkA rely on closed-source models like GPT-3.5/GPT-4. Additionally, Patel et al. (2024) employ Qwen-1.5-72B, *which is ten times larger than our model*. For results on models within the sub-10B scale, we direct nEkA to the WebArena leaderboard [https://shorturl.at/TFf5q]. Notably, except for AutoWebGLM, which uses extensive human demonstrations, all methods in the sub-10B category achieve less than a 7% success rate. In contrast, our model achieves a 7.2% success rate, establishing a new state-of-the-art at this scale without human demonstrations.
>
> ### Effectiveness of NNetNav-6k
>
> We respectfully clarify a misunderstanding regarding the NNetNav-6k dataset. While NNetNav-6k consists of 6,073 demonstrations, each demonstration (instruction, trajectory) generates multiple training instances based on the trajectory length. Thus, NNetNav-6k results in over 77,000 training instances, as described in Section 2 (Lines 138-145). In response to nEkA’s suggestion, we have now extended our experiments to utilize the full NNetNav-6k dataset. Using the same hyperparameters, we fine-tuned LLaMA-8b-instruct and achieved a 10.3% success rate on WebArena, a significant improvement over the 7.2% reported in Table 1, where we used a smaller dataset for fair comparison with other baselines. ***This result surpasses previously published results for sub-10B models using synthetic datasets and, to our knowledge, establishes our model as the state-of-the-art among agents that do not use closed-source models (like GPT-4 or GPT-3.5), or human demonstrations***:
> * LLaMA-8b-NNetNav [ours]: 10.3% (6k demonstrations)
> * AutoWebGLM-7b (S1): 2.5% (240 demonstrations)
> * Synatra-CodeLLaMA-7b: 6.3% (30k demonstrations)
> * Patel et al. (2024): 9.36% (Qwen-1.5-72B). Moreover, our performance exceeds that of Patel et al. (2024), a prior work mentioned by nEka, who used a much larger Qwen-1.5-72B-Chat model:
>
> We believe that these new results affirm the effectiveness of our approach, and we hope this addresses nEkA’s concerns and reinforces confidence in our contributions.

---

> > ### Author Response · Authors · 2024-11-26
> > **Gentle reminder**
> >
> > Thank you for your review! Since the discussion period is wrapping up soon, we just wanted to check if our responses addressed your concerns and if there’s anything else we can help clarify.
> >
> > We have also updated the draft to reflect some of these changes. Thank you.

---

> ### Comment · Reviewer_nEkA · 2024-11-27
>
> Thank you authors for the detailed response, and I really sorry for the late reply due to my personal matter.
>
> **> Re: W1**
>
> Thank you for listing the difference between BAGEL and NNetNav. However, I think current writing emphasizes that demonstration-first data collection method is what you newly introduced in this paper (Section 3), and you compare it against instruction first methods. I believe, instead of those high level concept, the difference you listed in the response is the novelty of this paper, and you should mention the relationship to BAGEL explicitly before presenting your method. Otherwise, the contribution of BAGEL is unfairly ignored. For the hirerachical exploration method, you mentioned it from the efficiency perspective. However, I think it is unclear whether this efficiency contribute to the performance improvement (maybe somewhat inefficient method may achieve better results).
>
>
> **> Re: W2**
>
> I think that Web Agent is a highly application-focused research area. Withouht very convincing reasons (here, this should be related to a performance-cost tradeoff), I dont't think the method with sub-optimal performance and lower cost is prioritized against the method with optimal performance and higher cost.
>
> For the baselines, Patel et al. (2024) employ Qwen-1.5-72B, but also apply LoRA for the finetuning. In contrast, I guess you employ full-parameter finetung. Because both two are still tractable approach to some extent and LoRA v.s finetuning causes performance tradeoff, this discussion may not provide convincing evidence.
>
> **> Re: W3**
>
> Thank you for the clarification on the dataset. For clarity, you may benefit from chaning order of section 6 into forward.

---

### Official Review · Reviewer_YpQW · 2024-11-03

**Soundness:** 2
**Presentation:** 2
**Contribution:** 1
**Rating:** 5
**Confidence:** 3

**Summary:**

This paper focuses on the problem of building an LLM based agent that can follow instructions by executing some actions (e.g. to perform web tasks). To achieve good instruction grounding this work focuses on collecting demonstrations. However, since expert demonstrations from humans can be expensive, they focus on using an exploration policy (an LLM) to generate action data. Heuristics are used to verify if the actions generated by the LLM indeed follow the true instructions. Once some data is collecting, hindsight relabeling is performed to re-label the original instruction (\hat{g}) and a reward is assigned. This data is then used to train a smaller LM and improve its performance using SFT. Experiments are shown on two domains. However, in  one of the domains (MiniWoB++) . On the harder (WebArena) domain, the approach shows  some improvements compared to a zero-shot baseline (however, this is a very weak baseline).

**Strengths:**

The paper is easy to follow and overall does a decent job at grounding different things. The paper also releases the demonstrations they have collected using the propose approach (although with a different LM than used for the paper).

**Weaknesses:**

The overall methodological contribution seems quite heuristic driven. For instance, the approach relies completely on \pi_explore, which is almost similar to the zero-shot baseline.  In scenarios where \pi_explore proposes decent actions this is probably reasonable, but if the environment is challenging then the exploration policy can be quite bad (e.g. if many different actions are quite similar). In this scenario, \pi_explore will not be able to collect any good high quality data.

There are other parts of the algorithm that are quite heuristic driven.  For instance, state changes between transitions are summarized using an LM (\delta_t) (this can be noisy (e.g.  the LM summary misses important details). This is probably not a huge problem for the datasets being considered but do the authors think that this can a potential concern which makes this approach hard to scale.

*Human demonstrations:* While human demonstrations are expensive, they can be much more higher quality. I think using a limited number of human demonstrations would be an important baseline for this work. Additionally, such limited number of human demos can also make the \pi_explore much better, which can potentially result in more higher quality data (compared to an off-the-shelf model).

*Why not train on the larger set of demonstrations?*  The authors have collected and are releasing a larger set of demonstrations but the training demos used in the paper are much smaller, why not use all the demonstrations and then compare the result? Also, it would also be interesting to see how the performance of the model scales as more demos are provided for SFT.

*Comparison to NNet (ablation baseline):* The ablation baseline is quite interesting, it basically takes just the instructions generated from this approach. These instructions are then used zero-shot with another LM to generate trajectories that will be used for SFT. From a results perspective, this approach is quite competitive with the proposed approach (e.g. only 1.2 points delta from proposed approach on WebArena). This is also when no data filtering was performed on this baseline.  Did the authors further explore and try to improve this baseline? Also, it would be interesting to see how the rollouts from this baseline differ from the proposed approach.

**Questions:**

please see above

---

> ### Author Response · Authors · 2024-11-18
> **Thanks for your review + some new experiments with NNetnav6k**
>
> We sincerely thank YpQW for their review. Below, we address some comments:
>
>
> ### Using heuristics:
> We would like to respectfully draw YpQW's attention to more mature literature on self-supervised exploration, where collecting synthetic demonstrations using exploration policies is a well-established practice. We agree that in particularly challenging environments, $\pi_\text{explore}$ may struggle to gather meaningful trajectories. We also acknowledge that our LM summarizer could potentially overlook important details when comparing differences. However, for websites, where the task is to summarize differences between two accessibility trees, we have found that this method works effectively.
>
> Collecting data in a completely unsupervised manner, particularly in difficult sequential decision-making environments, remains an open problem. Our paper specifically addresses the collection of demonstrations on websites where our practical heuristics have proven effective. We do not claim generalization to other types of grounded environments.
>
> ### Using human demonstrations:
>
> We agree that incorporating human demonstrations is an interesting avenue but falls outside the current scope of this work. To date, there is no large-scale dataset of human demonstrations for complex web-based tasks. Additionally, collecting such demonstrations at the same scale as NNetNav would be significantly more resource-intensive, as it requires training annotators with domain-specific knowledge (e.g., GitLab requires understanding the basics of version control software).
>
> ### Why not train on the larger set of demonstrations:
>
> Thank you for this suggestion. We have now extended our training to include the full NNetNav-6k dataset. As detailed in Section 2, we convert the 6,000 demonstrations into over 77,000 training instances, as each demonstration generates multiple training examples based on trajectory length. Using the same hyperparameters, we fine-tuned LLaMA-8b-instruct, resulting in a WebArena success rate of 10.3%—a significant improvement over the 7.2% reported in Table 1 (where we used a smaller dataset for fair comparison with other baselines). ***This represents substantial progress over previously published results for sub-10B models using synthetic datasets and, to our knowledge, establishes our model as the state-of-the-art among agents that do not use closed source LLMs (like GPT-4 or GPT-3.5), and with no additional human supervision***:
> * LLaMA-8b-NNetNav [ours]: 10.3% (6k demonstrations)
> * AutoWebGLM-7b (S1): 2.5% with 240 demonstrations
> * Synatra-CodeLLaMA-7b: 6.3% with 30k demonstrations
> * Furthermore, our results exceed those of Patel et al. (2024), a prior work mentioned by reviewer-nEkA, who used a significantly larger Qwen-1.5-72B-Chat model: Patel et al. (2024): 9.36% with Qwen-1.5-72B
>
> We hope these updated results more convincingly highlight the downstream utility of NNetNav-6k, demonstrating not only its uniqueness as the first dataset of complex instructions on WebArena websites but also the effectiveness of our approach. We believe these findings will strengthen your confidence in the quality and impact of our dataset.
>
> Comparison to NNetNav (baseline): To clarify, both SFT (NNetnav + distil.) and SFT (NNetnav) are methods proposed in our work, so we are neutral regarding which performs better. However, SFT (NNetnav + distil.) is more resource-intensive, requiring twice the environment interactions and an additional round of LLM sampling per instruction. We agree that further exploration of rollout differences between NNetNav trajectories and instruction-following ones would be insightful. We have included preliminary experiments on this in Lines 354-366 of Section 5. If YpQW has specific suggestions, we would be happy to conduct and report additional experiments.

---

> > ### Author Response · Authors · 2024-11-26
> > **Gentle Reminder**
> >
> > We thank you again for your feedback on our work, We wanted to check in if our provided responses address your concerns, and see if there are any further questions that we can help address.
> >
> > Thanks!

---

### Official Review · Reviewer_ezyY · 2024-11-04

**Soundness:** 3
**Presentation:** 2
**Contribution:** 3
**Rating:** 6
**Confidence:** 3

**Summary:**

This paper considers the task of instruction following for web browser agents, and proposes an approach for synthetic data generation to create data sets that can be used to fine tune LLM-based web browser agents. The authors make the argument that current "instruction-first" approaches to synthetic data generation are inadequate, and propose an "interaction-first" approach to data generation in which a sequence of website interactions is generated and then retroactively labeled to enable fine tuning. The authors' empirical results suggest that this kind of data generation leads to improvements on some standard benchmarks for browser agent performance. The paper also introduces a large-ish (several thousand data points) fine-tuning data set generated with the proposed retroactive labeling method.

**Strengths:**

Overall the paper has many strengths. The target task and range of applicability for the proposed method are clearly articulated, and the core idea that is proposed ("retroactive" or "interaction-first" labeling) is very clever. The empirical results from the fine-tuning experiments seem reasonably good overall, though see discussion below in weaknesses for some points where fine tuning results could be clarified. It is clear that effort has been put into making the work reproducible - the prompts used for each of the system components are available in the appendix, and this helps to understand what's going on even where descriptions in the paper could be clearer. The writing is, on balance, reasonably clear and free from distracting problems related to grammar or diction.

Overall, I really like the idea of this paper, and feel like it could be broadly useful for synthetic data generation in many contexts.

**Weaknesses:**

I am inclined to believe the core claims of the paper, and I think the ideas of the paper are quite good. I have a number of issues with the presentation of the work.

- Clarity. There are a few points where the paper could be clearer. For instance, the introduction's discussion of your pruning approach (line 096) could be clarified, since trajectories aren't yet defined, and trajectory prefixes as the basic unit of labeling is not necessarily an obvious choice. More significantly, on line 125 you should clarify that it is the elements of the observation and action spaces that have string representations, rather than the spaces themselves. This becomes clear from context as you read the paper, but it would be better to just say what you mean as you introduce the formalism. At several places you use the word "plausible" (for example on line 198) to refer to the generated instructions, but as far as I can see you don't define what "plausible" means or attempt to quantify it. Given that the quality of the generated instructions is critical to the viability of your approach, it surprised me that you did not put more effort into directly evaluating those instructions. I would guess that the indirect metric of "how good are the models that are trained on this data" is an acceptable proxy for instruction quality, but I'm sure you would strengthen your argument with some more direct assessment (of some kind; I concede that how best to do this is not obvious).

- Notation. Your notation in Sections 2 and 3 looks pretty textbook for RL-type applications, but I'm not sure it entirely represents the task you're working on. For example, it appears from line 124 that $\pi$ is a probability distribution over actions, but then line 129 makes it clear that $\pi$ is in fact a probability distribution over strings. This is also hard to interpret consistently with the prompts from the appendix, which make it seem like you are interested in strings of text sampled from models rather than the probabilities of these strings. I know that you need to include a certain amount of mathematical notation to get papers accepted, and that ultimately this notation is not really necessary for understanding the contributions of the paper, but if you're going to use the notation it would be helpful if some of it were more clearly defined. For instance, I would specify whether $\mathcal{A}$ is a finite set of possible actions (as one might guess looking at listings 1 and 8 in the appendix), or if $\mathcal{A}$ is a much larger set of possible outputs from an LLM. Then at a minimum I would specify whether the policies are working directly with probabilities or with strings, and whether those strings are drawn from a giving finite set or if they can be freely generated by some model.

- Reward model. In the text of the paper (line 199) it seems like you use binary rewards. This is not consistent with the prompts for $s_{\text{LM}}$. I think you clarify this on line 344, but your use of the same notation for the reward models that are binary and graded is slightly confusing.

- Results. Cross website generalization looks surprisingly weak, which is a little troubling for your main claims. Also, the results tables would benefit from a careful review. In particular, make sure you have units on all your numbers, either in the table itself or in the caption or in the main body. For instance, the Table 1 caption is good, but the captions for Tables 2, 3, 4 are not necessarily clear. In particular, with Table 4 and the discussion on lines 415 to 417 I'm not convinced that increases of 0.06 and 0.13 are significant because I don't entirely know what units those numbers have. Is the range of reward here 0 to 1 or 1 to 5? I believe that this can be clarified in the text of the paper, and that you should do so.

- NNetNav-6k. The major issue here is that I'm not clear whether or not you did anything with this data set other than release it. It's not clear that "diversity of verbs" or "instruction length" have any bearing on whether or not the dataset is any good, and it doesn't look like you've used the data set to fine tune any models (if this is not correct you should emphasize your use of the 6k data set more in the main body of the paper). Overall there's no assessment of quality for the data set as far as I can tell. I'm not sure that mentioning this data set adds anything to the paper other than length, given the current discussion of the data set in the paper. I fully believe that the data set would be useful, so if there's any way to express that in the main body of the paper you should do so.

Less importantly, I'm surprised that you found that GPT-4o and Llama-3 had "similar quality" based on my experience with those models. This is the kind of claim that would benefit from more rigorous justification.

Also (and this is just a quibble): I strongly doubt that web browsing is, in general, a deterministic process. But again, I think you're probably only saying that to introduce some notation.

**Questions:**

Biggest question: are you claiming novelty of the retroactive labeling approach? If so, I would emphasize this more in the paper because it's a good idea. If you are not claiming novelty, I would point out the related work more clearly.

Have you considered retroactive labeling in domains beyond browser agents? The idea feels generally applicable. Maybe worth a mention in the conclusion, since the current conclusion is just a laundry list of things you've done.

**Details Of Ethics Concerns:**

No ethics concerns.

---

> ### Author Response · Authors · 2024-11-18
> **Response + new experiments establishing SoTA**
>
> We thank ezyY for their thoughtful review. We appreciate your positive feedback on the ideas presented in our paper. Below, we address your concerns with clarifications and updates:
>
> ### Notation and clarity
>
> Thank you for your comments on notation. In addition to fixing the draft, we would like to provide a few clarifications here:
> * Defining Plausible: We appreciate the suggestion for a clearer definition. In our context, we use the term "plausible" to refer to instructions that align with the genre of a given website (e.g., a shopping instruction on an e-commerce website or a "comment on a post" instruction on a social media site).
> * Quality of generated instructions: This is an excellent point. We use Figures 3 and 4 to illustrate aspects of diversity, while Table 5 provides specific examples of our generated instructions. Additionally, we have included new results obtained by fine-tuning LLaMA-8b on NNetNav data, which further supports quality claims. We agree that evaluating instruction quality comprehensively remains an open challenge, and we appreciate your acknowledgment of this complexity.
> * About $\mathcal{A}$: We apologize for any confusion related to this notation. Based on your suggestion, we will simplify it in the revised version. The action space of the LLM is indeed finite but quite large, as many actions refer to an accessibility tree ID, with some observations containing up to 1,000 such IDs.
>
> ### Reward model:
>
> We apologize for the typo and any resulting confusion. To clarify, we employ two distinct reward models:
> * The first is a binary reward model used for NNetNav data creation (referenced in Lines 199-200).
> * The second is a model-based evaluation (described around Line 343) using GPT-4v, which assigns scores from 1-5 based on a different prompt. This model is solely used for evaluation. Notably, there is a typo in Line 344, where we mistakenly referred to Appendix A.2 instead of the correct Appendix B, which contains the prompt for this model-based evaluator.
>
> ### Cross-website generalization:
>
> We would like to clarify the context of our cross-website generalization experiment. Our intention here is not to demonstrate the effectiveness of NNetNav but rather to explore the generalization capabilities of the base LLM. Specifically, we split our data to train on NNetNav demonstrations from Shopping, Maps, and CMS, and then assess generalization to instructions on unseen websites. While NNetNav can be applied to any website, this section aims to evaluate how well current language models generalize from demonstrations on specific websites to new ones.
>
> ### Using NNetnav-6k:
>
> Thank you for your comments. We have now trained language models using the entire NNetNav-6k dataset. As outlined in Section 2, we convert the 6,000 demonstrations into over 77,000 training instances since each demonstration results in |trajectory| training examples. We fine-tuned LLaMA-8b-instruct with the same hyperparameters as presented in the paper and Appendix D, achieving a WebArena success rate of 10.3%. This represents a substantial improvement over the results in Table 1 (7.2%), which focused on a smaller dataset for fair comparison with other baselines. **Notably, this model is the state-of-the-art among approaches that do not use GPT-4 / GPT-3.5, and use no additional human supervision**:
> * AutoWebGLM-7b (S1): 2.5% with 240 demonstrations
> * Synatra-CodeLLaMA-7b: 6.3% with 30k demonstrations
> * Patel et al. (2024): 9.36% with Qwen-1.5-72B. Furthermore, our results exceed those reported by Patel et al. (2024), which used a much larger Qwen-1.5-72B-Chat model.
>
> We hope these updated results more effectively demonstrate the downstream utility of NNetNav-6k, beyond being the first and only dataset of complex instructions on WebArena websites.

---

> > ### Author Response · Authors · 2024-11-26
> > **Gentle reminder**
> >
> > Thank you again for your detailed feedback on our work! Since the discussion period is wrapping up soon, we just wanted to check if our responses addressed your concerns and if there’s anything else we can help clarify.
> >
> > Thanks!

---

### Note · Authors · 2024-12-14

I have read and agree with the venue's withdrawal policy on behalf of myself and my co-authors.